# Review of Magnetic Properties and Texture Evolution in Non-Oriented Electrical Steels

Yizhou Du, Ronald O'Malley  and Mario F. Buchely *

Department of Materials Science and Engineering, Missouri University of Science and Technology, Rolla, MO 65409, USA
* Correspondence: buchelym@mst.edu; Tel.: +1-573-341-6627

**Abstract:** Electrical steels can be classified into two groups: grain-oriented (GO) and non-oriented (NGO) electrical steel. NGO electrical steels are mainly considered as core materials for different devices, such as electric motors, generators, and rotating machines. The magnetic properties and texture evolution of NGO electrical steels depend on multiple factors (such as chemical content, heat-treatment, and rolling process) making the development of new products a complex task. In this review, studies on the magnetic properties of NGO electrical steels and the corresponding texture evolution are summarized. The results indicate that further research is required for NGO electrical steels to ensure high permeability and low core loss properties.

**Keywords:** electrical steels; twin-roll strip casting; magnetic properties; core loss; texture



## 1. Introduction

Electrical steels are widely used as core materials for a variety of electrical equipment, such as transformers, generators, rotating machines, etc. Due to rapid development in the manufacturing and automation sectors, applications for electrical steels are continuously expanding [1].

Electrical steels are commonly classified as grain-oriented (GO) and non-oriented (NGO) electrical steels. The main applications of GO electrical steel include transformer cores, power reactors, hydro-generators, turbo-generators, etc. Conversely, the main applications of NGO electrical steel include generator cores, electric motors, electrical meters, etc.

In the case of NGO electrical steels, the core loss and the magnetic induction are critical features that have been investigated in numerous studies [1]. In electric motor applications, efficiency and torque are the most focused properties [2]. Core loss is directly related to motor efficiency. The lower the core loss, the higher the motor efficiency [2]. Magnetic induction is directly proportional to motor torque. Figure 1a shows that, in practice, it is difficult to develop products with both high magnetic induction and low core loss.

Grain size, inclusions, precipitates, defects, and grain orientation of electrical steels have a distinct effect on magnetic properties. These parameters are influenced by multiple factors, such as steel composition, thermomechanical processing, and so on. Therefore, a review of each factor's influence on the texture evolution of NGO electrical steels would be a valuable contribution to the steelmaking community. Knowledge of these relationships will help engineers control the texture and magnetic properties in NGO electrical steel production processes.

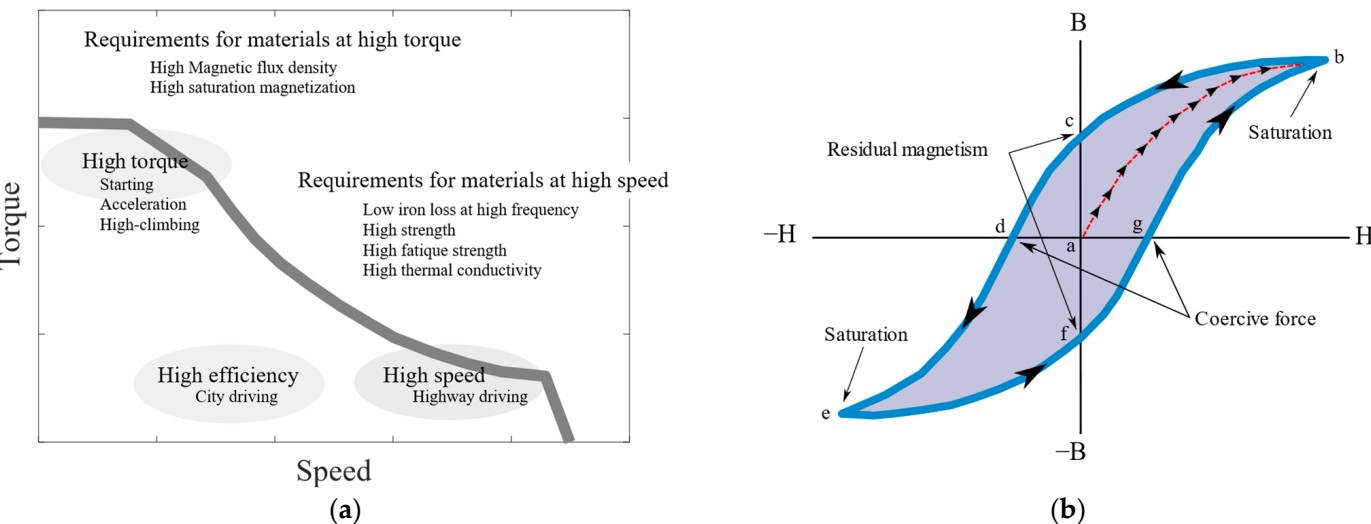

**Figure 1.** (**a**) Demands of core materials on an electric vehicle motor's applications, reprinted/adapted with permission from [2]; and (**b**) Magnetic flux density B against the magnetizing force H (B-H) curve with some critical points.

## 2. Magnetic Properties

As shown in Figure 1b, magnetization curves (B-H curves) are plotted by the values of magnetic flux density (B) against the field strength or magnetizing force (H). As the alternating current (AC) condition is continuously changing the direction of the magnetizing current through the coil in electrical steel applications, this alternating magnetic field can produce a magnetic hysteresis loop in the core material. Residual magnetism is the induction that is left in the material when it has already been magnetized. Coercivity force (or coercivity) is the magnetic field strength that is required to demagnetize the material when it has already been magnetized. When all the magnetic domains within the material have aligned with the external magnetic field, a saturation effect is observed. The slope of the B-H curve at any location is defined as the incremental permeability. Sometimes, permeability is measured from the origin to the target location. This slope is called apparent permeability. The area under the B-H curve is the hysteresis loss in each B-H cycle [3].

In studies addressing the magnetic properties of NGO electrical steels, core loss, coercive field, permeability, and magnetic induction are the main properties that dictate the magnetic material behavior. The important properties of electrical steel differ for alternating current (AC) and direct current (DC). For AC applications, core loss is of major importance, because the alternating nature of AC causes very rapid domain flipping, and thus hysteresis loss of the core loss becomes a larger component of the energy loss [1]. In contrast, permeability, coercive field, and magnetic induction properties are more important in applications under DC conditions. Hysteresis loss also occurs in applications under DC conditions; however, it is not the main influence on energy loss [1].

### 2.1. Magnetic Properties under AC Conditions

As mentioned previously, core loss is one of the most important magnetic properties for application under AC conditions. Under AC conditions, some power is lost in the core of the device, transforming it to heat or noise. This energy loss is called core loss. It is commonly accepted that core loss can be separated into hysteresis loss ($P_h$), anomalous loss ($P_a$), and eddy current loss ($P_e$).

$$P_{tot} = P_a + P_e + P_h, \tag{1}$$

$P_h$ is the energy loss that occurs during every cycle that the material undergoes an applied field change. In practice, hysteresis loss depends on the grain size, inclusions, precipitates, the presence of defects, the orientation, and the applied frequency.

$P_e$ represents the energy loss caused by the electrical current. This electrical current is induced by a change in the magnetic field. Eddy current depends on the chemical composition and geometry of the material. $P_e$ is aided by increasing the material's resistivity which is controlled by higher contents of Si and Al and thinner steel laminations. As shown in Figure 2, the use of thin laminations leads to a decrease in the eddy current loss for applications in an electric motor.

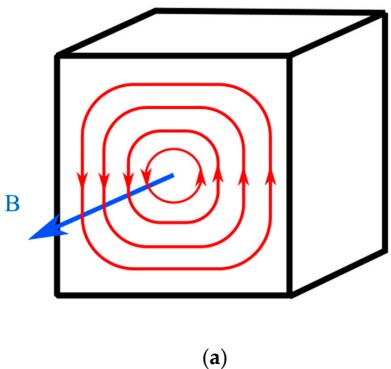

(a)

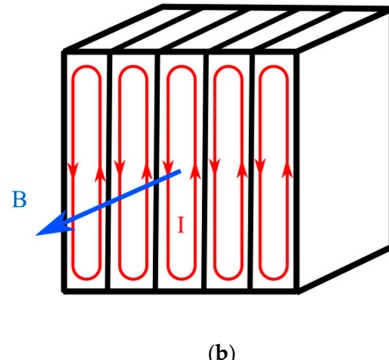

(b)

**Figure 2.** (**a**) Eddy current in the solid iron core. (**b**) Separating the iron core by thin laminations that are parallel to the field (to reduce the eddy currents, insulation layers were coated between laminations). In this figure, B is the magnetic field and I is the induced current. The direction of the arrow indicates the flow of the magnetic field.

$P_a$ is the energy loss present after the calculation of $P_e$ and $P_h$ (the eddy current and the hysteresis loss, respectively). Owing that it represents part of the energy loss that is not considered in the eddy current loss calculation in detail, the anomalous loss is also called the excess eddy current loss or excess loss [4]. The energy losses of $P_e$, $P_a$, and $P_h$ may be expressed as follows [5]:

$$P_e = \frac{\pi^2 B_{max}^2 t^2 f^2}{6\rho D}, \tag{2}$$

$$P_a = c_1 \frac{\sqrt{d}}{\rho} t^2 B_{max}^2 f^{3/2} \tag{3}$$

$$P_h = \frac{f}{D} \oint B dH \tag{4}$$

where $c_1$ is an experimentally determined constant, $d$ is the material grain size, $B_{max}$ is the peak magnetic induction, $t$ is the lamination thickness, $D$ is the material density, $f$ is the working frequency, and $\rho$ is the material resistivity.

The Equation (3) about anomalous loss expression is not universally accepted. G. Bertotti et al. [6] reported some different physical models on energy loss by expressing them in magnetic laminations under one- and two-dimensional fields. Other than that, there are also some works which compute energy losses based on the applications. A.J. Moses [7] reported an algorithm to estimate the iron loss of power transformers from quantification of the contributions of the effect of joints, rotational and harmonic flux, stress, interlaminar flux, and core geometry.

Taking into account the three energy losses that exert an influence on frequency, the $P_{tot}$ total energy loss may be obtained as follows [5]:

$$P_{tot} = k_a f^{3/2} + k_e f^2 + k_h f, \tag{5}$$

where $k_e$, $k_a$, and $k_h$ are the parameters of the eddy current loss, the anomalous loss, and the hysteresis loss, respectively. As expressed in Equation (4), $P_e$ and $P_a$ are influenced most by the applied frequency. In practice, $P_e$ has the largest contribution to $P_{tot}$ when the applied frequency is higher than 400 Hz, just like the e-mobility [5].

There are new challenges presented by application under high frequency like e-mobility. Other than the effect shown in Equation (4), a high frequency AC condition makes the number of effects apparent. Those effects include but are not limited to skin effect, proximity effect, and geometric effects. A. D. Podoltsev et al. [8] reported a numerical model for calculation of eddy current losses under high frequency conditions. It computes the leakage field, taking into consideration the effective magnetic permeability of the multiturn winding as a heterogeneous medium.

Magnetic induction $(B_H)$ is another important magnetic property for application under AC conditions. Magnetic induction of electrical steels at a given applied field critically depends on the microstructure and the present crystallographic texture. Gomes et al. [9] reported a general quantitative model for the dependence of the magnetic induction at a given applied field as a function of the mean grain size, a texture-related parameter, and the Si content of the material. This relation is expressed as follows [9]:

$$B_H\left(d, A, Si_{eq}\right) = p_0 + p_1 \times A + \frac{p_2}{D} + p_3 \times Si_{eq},$$ (6)

where $A$ is the texture parameter, $p_0$, $p_1$, $p_2$, and $p_3$ are material parameters, $d$ is the grain size, $Si_{eq}$ is defined as ($Si + 2 \times Al$) in wt.%.

### 2.2. Magnetic Properties under DC Conditions

For electrical steel, permeability is a measure of the resistance of a material in opposition to the formation of a magnetic field, while coercivity is a measure of the ability for a ferromagnetic material to withstand an external magnetic field without becoming demagnetized.

Permeability is defined by the instantaneous slope of the *B-H* curve, and thus is sensitive to induction [1]. For applications under low induction conditions (1 T), at 50 Hz, the grain size of NGO electrical steels has a strong influence on the corresponding permeability [10,11]. Grain growth during recrystallization affects the amount and distribution of desirable magnetic textures. Under high induction conditions (1.5 T), the effect of texture on permeability and magnetic flux density is more significant than that of grain size.

Furthermore, coercivity and permeability are inversely proportional [12,13]. It is widely known that a larger grain size results in higher permeability and lower coercivity, which is also the cause for a decrease in hysteresis loss. To elaborate, grain boundaries may delay and impede the movement of the domain wall. However, a larger grain size also leads to a larger domain size, which in turn increases the core loss, especially anomalous loss [14]. Thus, an optimum grain size results in a minimum core loss. De Campos [15] described the optimum grain size ($G_{sOp}$) as follows:

$$G_{sOp} = \left(\frac{c_2 \rho}{B^{2-q} t^2 f^{1/2}}\right)^{2/3},$$ (7)

where $c_2$ is an experimentally determined constant, $B$ is the magnetic induction, $\rho$ is the resistivity, $t$ is the sample thickness, and $f$ is the operating frequency. Steinmetz experimentally determined that $q = 1.6$ [16].

## 3. Effects of Si Content on NGO Electrical Steels

Silicon (Si) content of NGO electrical steels has a strong effect on magnetic properties [17]. Higher Si content increases the material resistivity, which leads to a decrease in the eddy current loss. Furthermore, the addition of Si content also led to a decrease in magnetocrystalline anisotropy [18].

Magnetocrystalline anisotropy is the property that defines when ferromagnetic material takes more energy to magnetize in some directions than others. Decreasing magnetocrystalline anisotropy leads to higher permeability. Magneto-striction and saturation induction are also lower when the Si content increases.

However, it is also observed that steel brittleness increases when the Si content is higher than 3 wt.%, which has a significant effect on cold deformability [19–21].

Shimanaka et al. [22] found that in NGO electrical steels, a higher optimum grain size is obtained as the Si content increases. For instance, 1.85 wt.% Si steels have an optimum grain size of approximately 100 μm, and 3.2 wt.% Si steels have an optimum grain size of approximately 150 μm [23].

*High Silicon Electrical Steel Ordering*

Brittleness of high Si steels depends on the grain size, ordered phase structure, and grain boundary impurities [24–26]. As shown in Figure 3a, when the Si content of the steel increases, $A_2$, $B_2$, and $D0_3$ ordering are observed. $A_2$, $B_2$, and $D0_3$ are the Strukturbericht symbols that designate different crystal ordering structures. These structures are best explained by the superlattice structure, as presented in Figure 3b. In Fe-Si alloys, two types of ordering phases were studied: $B_2$ structure (Fe-Si) and $D0_3$ structure (Fe$_3$Si ordering type) are formed with a Si content of 5.3–11 wt.% [27]. $A_2$ represents the simply $\alpha$-Fe BCC structure.

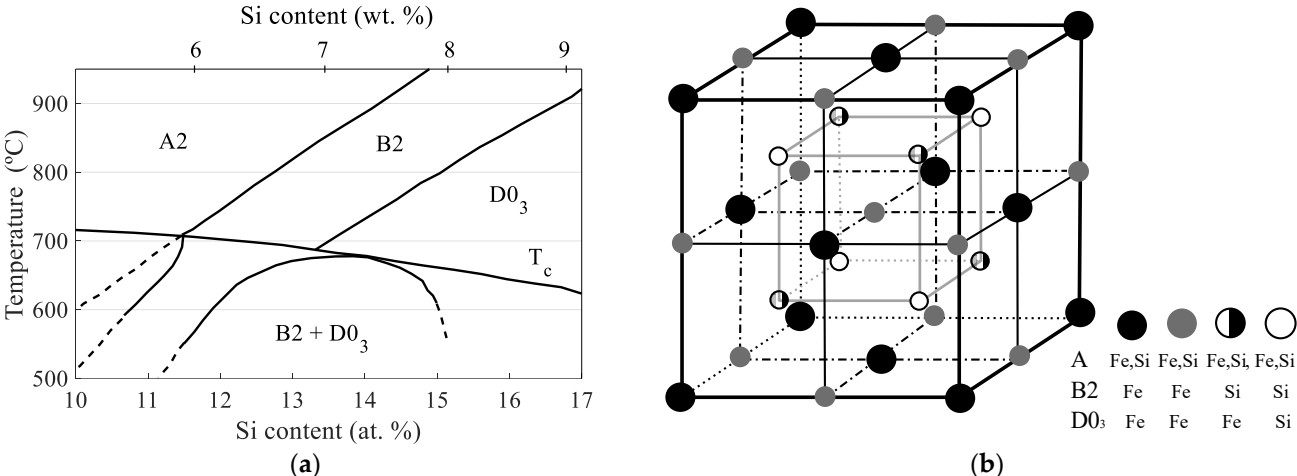

**Figure 3.** (**a**) Section of the binary phase diagram of the Fe-Si alloy system [20]. (**b**) Superlattice structures of A2, B2, and $D0_3$ structures in high Si electrical steels. A2 is disordered, allowing Fe and Si to allocate in any available site. In B2 ordering, solid gray and black dots represent the sites where Fe is present, whereas gradient and open dots represent the sites where Si is observed. In the $D0_3$ texture, open dots represent the sites where Si is located, while Fe were located on other sites [21].

X-ray diffraction (XRD) analysis has shown that $D0_3$ has a special superlattice peak, which corresponds to {111} planes. $B_2$ and $D0_3$ {100} planes share a {200} superlattice peak [1,28]. Dislocations interfere with these orderings and cause a strengthening effect [29]. Superdislocation slip deformation may affect mechanical properties. This deformation mechanism is also observed in $B_2$ and $D0_3$ lattices in high Si electrical steels [30]. In some studies, $B_2$ and $D0_3$ ordering increased the magnetic properties. The growth of $B_2$ leads to a higher specific magnetization, while the growth of $D0_3$ leads to a low coercive force and maximum permeability [31].

Rapid cooling at a critical cooling rate can suppress $D0_3$ ordering and reduce the size of $B_2$ ordering in Fe-Si alloys [32]. An exponential relationship between the Si content and the critical cooling rate can be observed in Figure 4 where A1 to A5 are air quenched, B1 to B3 are oil quenched, C1 to C3 water quenched, and D1 to D4 are brine quenched. In another study, 5–6 wt.% Si 2 mm thick strip samples were hot-rolled at 1200–900 °C and cooled by air. The strip samples were then annealed at 850 °C for 1 h and cooled at different cooling rates (as shown in Figure 5). As seen in this figure, the high cooling rate can reduce the domain size of $B_2$ phase and suppress the $D0_3$ phase [33]. In addition, deformation decreases high Si ordering and reduces brittleness [34].

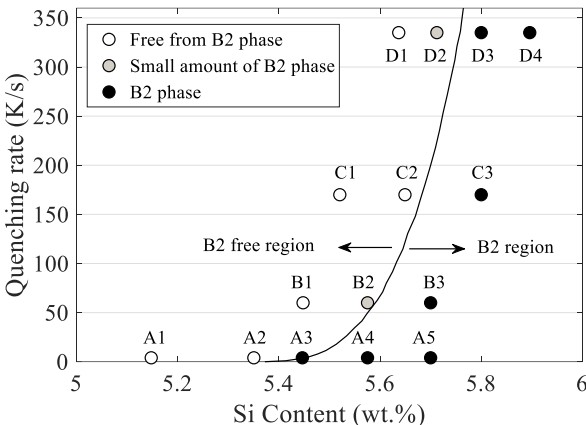

**Figure 4.** Si content with respect to critical cooling rates. The samples were annealed at 850 °C for 1 h, and then quenched at different cooling rates [20].

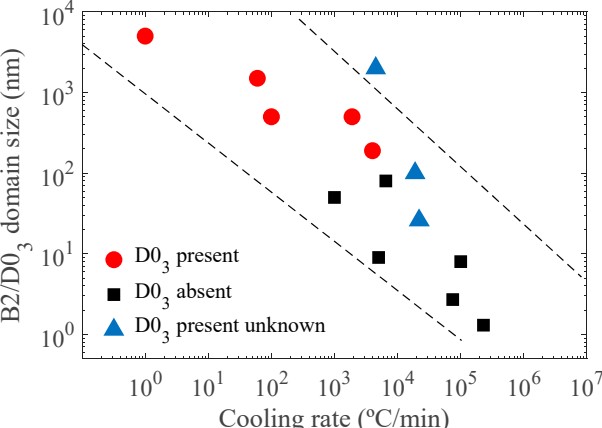

**Figure 5.** B2/D0$_3$ domain size regarding critical cooling rates [33].

## 4. Effects of Thermomechanical Processing

Thermomechanical processing (TMP) plays an important role in the final microstructure and magnetic properties of electrical steels [35–40]. TMP involves reheating, hot-rolling, cold-rolling, and an intermediate and final annealing (recrystallization annealing) process, which influence the inclusion distribution, microstructure, and texture, which in turn influence the final magnetic properties.

The product's final magnetic properties (after final annealing) are directly affected by the texture distribution [41]. The texture of electrical steels is influenced by the following factors: (1) steels chemical composition; (2) possible $\gamma$ to $\alpha$-phase transitions during cooling; (3) high solidification cooling rates; (4) directional solidification; (5) recrystallization annealing after the cold-rolling process; (6) deformation regime [42]; and (7) magnetic annealing [43].

Rolling temperature has a significant influence on texture and microstructure evolution [38]. The texture produced by hot-rolling also affects the texture and microstructure evolution during the cold-rolling process [44]. Nuclei orientation and the growth rate of these nuclei influence the recrystallization texture. There are two main theories that describe texture development, one of which is the nucleation-oriented theory. This theory assumes that nuclei that have a specific orientation grow rapidly. The fast growing nuclei orientation affects recrystallization texture [40]. The second theory is a growth-oriented theory, which assumes that there are some specific orientation relationships for which grain boundaries migrate more rapidly [17].

Euler angles are the three rotations that align the <100> direction with the rolling direction. The orientation distribution function (ODF) in Figure 6 shows the sections when one Euler angle (Bunge notation) $\varphi_2$ is at (a) 0° and (b) 45°.

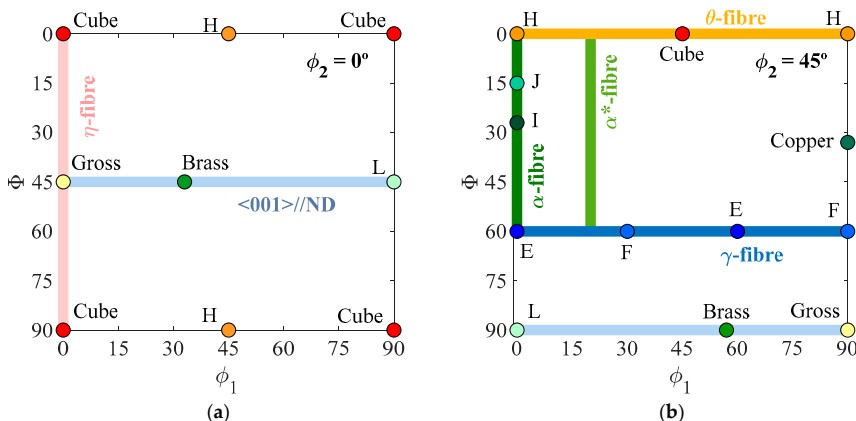

**Figure 6.** Fiber structures and orientations in electrical steels (Bunge notation) at: (**a**) φ2 = 0° section, and (**b**) φ2 = 45° section.

The notation of texture {110}<001> represents the planes of the form {110} that are parallel to the surface of the sheet, and directions of the form <001> that are parallel to the rolling direction [45]. Rolling textures are often represented by "fibers" instead of specific orientations.

During the thermomechanical processing of electrical steels, some specific textures are observed: <111> //normal direction [ND] (γ-fiber), <100>// [ND] (θ-fiber), {hhl} $< \frac{h}{l} + 1\frac{h}{l} + 2\frac{h}{l} >$ (α*-fiber), <110> // rolling direction [RD] (α-fiber), and <100>// RD (η-fiber). The orientation and fiber structures are shown in Figure 6 [46].

In electrical steels, α-fiber, α*-fiber, and γ-fiber are commonly formed during thermomechanical processing [47]. This is caused by the oriented nucleation behavior at elongated deformation bands. The γ-fiber and α-fiber deformation bands are formed by the rotation mechanisms in the hot- and cold-rolling processes [35,47,48]. For electrical steels used in rotating magnetic field (electric motors), λ-fiber is the ideal texture for good magnetic properties because it has the best magnetization direction <001> in the plane of the sheet. In this case, any {hkl} with <001> axes in the plane of the sheet is preferred. For magnetic circuits, the best texture is η-fiber, since this makes it possible to orient the magnetic flux along the <100> direction for almost its entire path.

The γ-fiber is not favorable for magnetic properties of electrical steels [49]. In conventional steel processing, γ-fiber evolves preferentially after annealing [50]. The recrystallization process occurs by consuming λ–fiber and α-fiber texture grains. This phenomenon is caused by the difference in orientation-dependent stored strain energy, which provides the activation energy for γ-fiber recrystallization [51,52]. Thus, it is difficult to weaken the γ-fiber texture and strengthen the λ–fiber texture. However, a secondary recrystallization step has proved helpful in controlling the γ-fiber texture in the annealing process, forming a strong rotated cube texture [40,53,54].

Because γ-fiber texture grain tends to nucleate at the grain boundary [47,55], γ-fiber texture formation is decreased by controlling the grain boundary nucleation environment [54]. Park et al. reduced the number of γ-fiber nuclei at the grain boundary by increasing the grain size [56]. Cunha et al. decreased the stored stain energy of the γ-fiber texture by two-stage cold-rolling [57]. Further, γ-fiber texture recrystallization is controlled through the temper-rolling process [58,59]. The development of shear bands has a significant influence on the formation of desirable recrystallization textures [60,61]. By temper-rolling at a temperature where dynamic strain aging (DSA) occurs, the formation of shear bands was increased [62]. If temper-rolling is performed above the DSA temperature, this increases the stored strain energy [59].

Other than the rolling process, the slightly deformation regime (cutting clearance) also has significant influence on the subsequent thermomechanical process. Higher cutting clearance can lead to fine-grained microstructure and generate a higher core loss. The anomalous loss is found to be the most sensitive energy loss to the cutting clearance [42].

## 5. Effect of Other Elements

### 5.1. Influence of Mn and S Content

From the metallurgical processing of electrical steels, S and Mn are usually present in the product. In some cases, Mn is one of the major components in the NGO electrical steel. During the production process, S and Mn may form MnS inclusions that have significant influence on the product properties.

The thermomechanical behavior of MnS precipitation has been thoroughly investigated [63–65]. It is widely accepted that MnS particles in NGO electrical steels have a higher likelihood of precipitating on the dislocations and grain boundaries [66,67]. MnS can also form interdendritically during solidification. Here the solidification rate can influence the size distribution of the MnS inclusions through the effects of the cooling rate on secondary dendrite arm spacing [68].

The precipitation and coarsening of MnS particles not only influences the core loss, but also affects magnetic aging in NGO electrical steels [69]. During heat treatment processes, when the precipitate size is close to the size of the magnetic domain, it will highly hinder the magnetic domain movement [70]. This may lead to an increase in coercivity and hysteresis losses, which is called the pinning effect. When the size of the precipitates is dimensionally similar to the thickness of the domain wall, the magnetic domain movement may be delayed [71–73]. Due to the thickness of the domain wall, only small inclusions have a significant effect on the magnetic properties. In practical production, a high Mn/S level chemistry is sometimes used to avoid this pinning effect. Driving the MnS level too low can easily form smaller MnS inclusions in a size range that is exceptionally detrimental. By comparison, large oxides and sulfide precipitates have very little influence on magnetics. In some cases, the fine inclusions that are nucleated in the solid state are the deleterious ones.

Ren Q et al. [74] reported the application of rare-earth elements (REMs) on the coarsening of sulfides in NGO electrical steels. The cast slabs (1.0 wt.% Si, 0.42 wt.% Al, 0.005 wt.% C, 0.004 wt.% S, and 0.004 wt.% REMs) were hot-rolled to 2.6 mm, normalized at approximately 900 °C for 3 min, cold-rolled to 0.3 mm, and finally continuous-annealed at 900 °C. A proper REMs addition can lead to an increase in the average grain size, decrease of micro-sized inclusions, fine MnS, optimization of recrystallization textures, and better magnetic properties.

Ca additions have also been used to decrease the number of small MnS particles. A 3.2 wt.% Si + 0.4 wt.% Al steel was 1-stage cold-rolled to 0.5 mm, and finally annealed at 900 °C. The effects of Ca on the core loss are illustrated in Figure 7 [75].

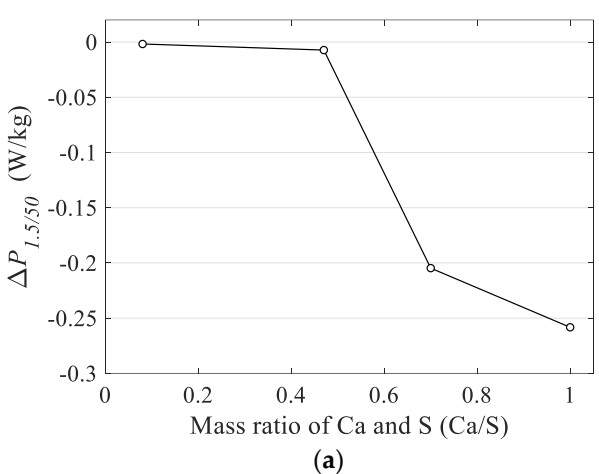
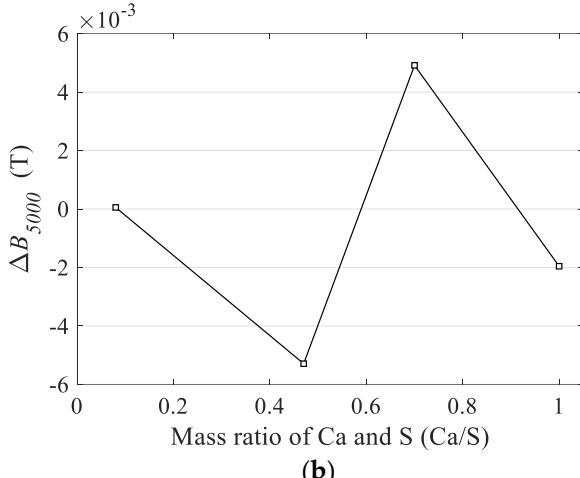

(**a**)　　　　　　　　　　　　　　　　　　　　　　　　(**b**)

**Figure 7.** Influence of mass ratio of Ca and S (in wt.% Ca/S) on final annealed strip. (**a**) Core loss, (**b**) magnetic induction [75].

### 5.2. Influence of Al Content

High Al content in NGO electrical steels may suppress grain growth. A comparison of the optical microstructures obtained is depicted in Figure 8. This grain growth suppression is caused by the pinning effect of AlN and other inclusions. This pinning effect is influenced by inclusion size. Furthermore, changes in annealing time or Al content resulted in no significant effect on the textures obtained [1], which may also be attributed to the pinning effect of inclusions.

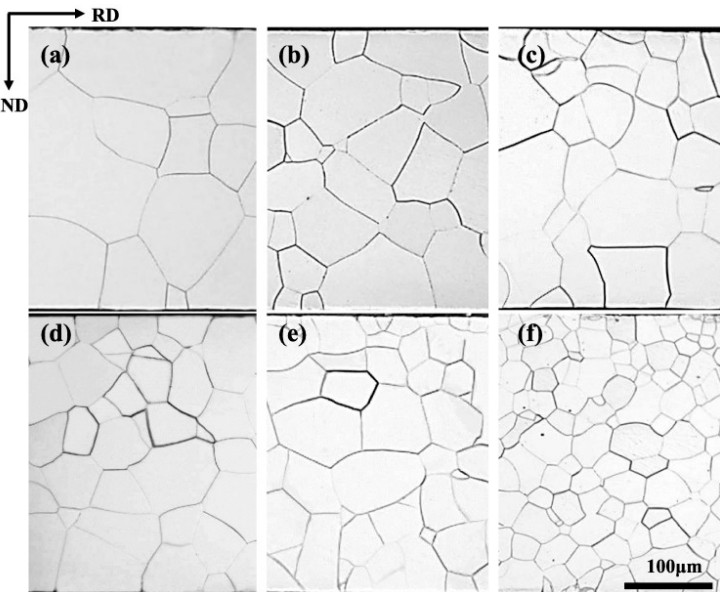

**Figure 8.** Optical microstructure of Fe-Al TD section samples with different Al content: (**a**) 0.53 wt.% Al, (**b**) 1.86 wt.% Al, (**c**) 2.64 wt.% Al, (**d**) 4.68 wt.% Al, (**e**) 6.54 wt.% Al, and (**f**) 9.65 wt.% Al after annealing at 1000 °C for 5 min [5].

When referring to magnetic properties, higher Al content may increase the $P_h$ of the total core loss because of the influence of the magnetic domain structures. On the other hand, higher Al content may also decrease the $P_a$ of the total core loss. The change in core loss has been ascribed to an increase in resistivity and the complexity of domain structures [1].

In practical production, high Al additions are used because they can force large nitrides to form during slab casting and reheating thereby rendering them inconsequential. AlN is well known to aid in the formation of high hkl textures such as {111} for drawing steels by pinning grains in a specific growth direction. Thus, some efforts to coarsen AlN particles are employed to limit their influence. It is also imperative that N be tied up in order to avoid magnetic aging [69]. However, it is extremely important that carbon is not tied up as carbides, because these are extremely problematic for magnetic properties.

### 5.3. Influence of B Content

The influence of B content in NGO electrical steels is correlated to the Al and nitrogen content. Lyudkovksy reported that over the composition range of 0.033–0.053 wt.% Al, the addition of 0.0007–0.0038 wt.% B could lead to BN precipitation instead of AlN [76]. This may increase the grain size and lead to an ideal texture. In the range of 0.0075–0.053 wt.% B content for Fe-6.5 wt.% electrical steels, Kim observed that an increase of B content might also have a grain-refining effect, and also improve bending strength and ductility at room temperature [77]. The influence of B in Fe-1.35 wt.% electrical steels has also been studied [78], as illustrated in Figure 9. As shown in Figure 9a, with the increase of B content up to 0.004 wt.%, the grain size increased. When the B content is higher than 0.004 wt.%, the correlation is inverted [70]. As shown in Figure 9b, the correlation between core loss, flux density, and B content also changed at 0.004 wt.% B content. The best magnetic properties were achieved with 0.004 wt.% B content, with core loss 3.616 W/kg and magnetic induction 1.792 T [70].

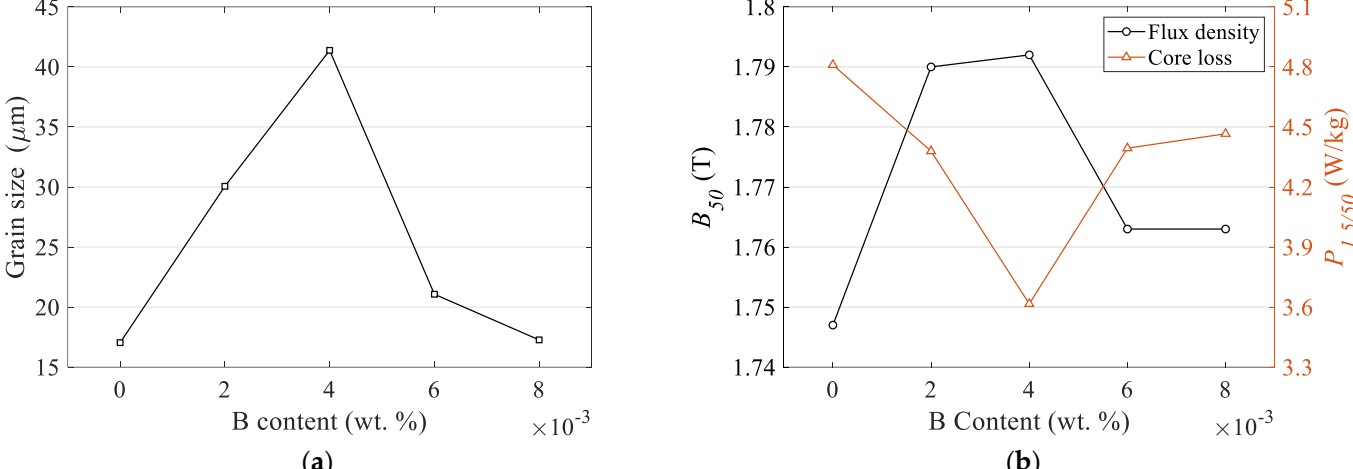

**Figure 9.** Boron (B) effect on annealed cold-rolled steel sheets: (**a**) grain size, (**b**) flux density and core loss [77].

The opposite effect (at a B content higher than 0.004 wt.%) in electrical steels may be caused by boron segregation at austenite grain boundaries. This segregation decreases the boundary energy and delays the $\gamma \rightarrow \alpha$ phase transformation in compositions that are not fully ferritic. It also promotes a favorable texture during the recrystallization process [79,80]. Boron is unique and, in contrast with AlN, hurts drawing textures and in some cases may be beneficial to magnetic properties. However, boron forms carbides which are difficult to control in processing.

### 5.4. Influence of Ce and Nb Content

It is widely accepted that the deoxidization and desulfurization function of Ce may coarsen inclusions, and thus lead to a decrease in the number of inclusions. Takashima et al. stated that in NGO electrical steels, Ce and Al have a significant influence on grain size during stress relieving annealing of NGO electrical steels [81]. Hou and Liao have observed that Ce also affects the texture evolution of the material [23]. The relations between cerium on the intensity of <100>//RD texture and <111>//ND texture are shown in Figure 10.

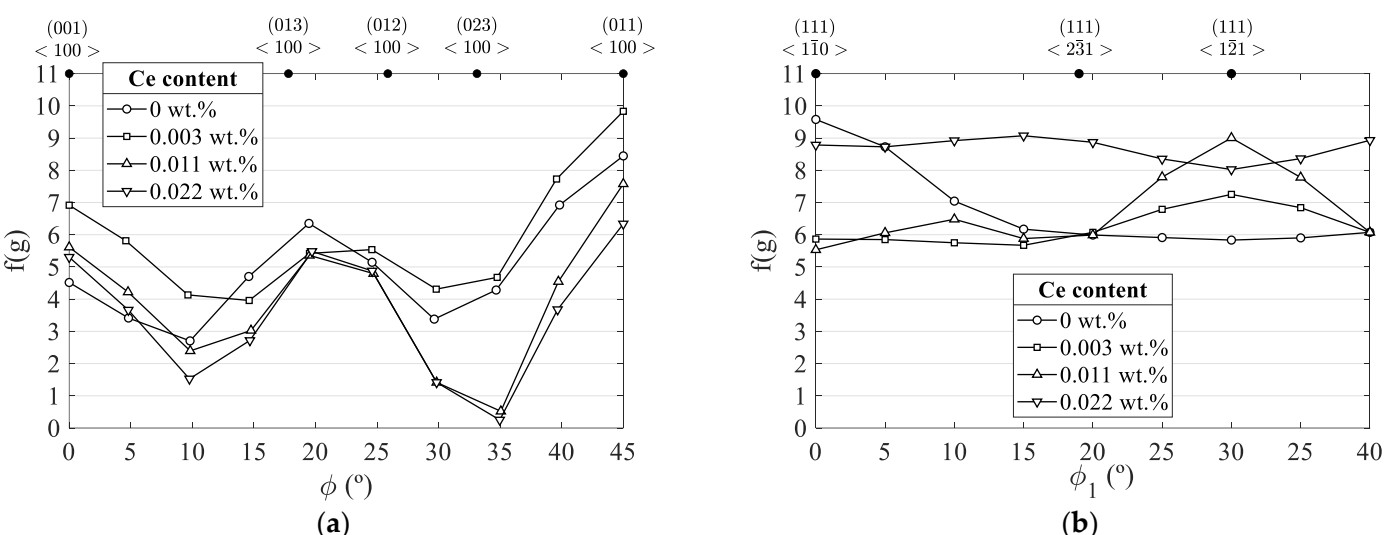

**Figure 10.** Effect of Cerium (Ce) on the intensity of (**a**) <100>//RD texture, and (**b**) <111>//ND texture. Hot-rolling finishing temperature: 890 °C; and hot band annealing temperature: 700 °C [23].

Based on the electronegativity difference of Nb, Fe, and Si, Nb-rich precipitates will destroy the ordered rearrangement between adjacent Fe and Si atoms in the matrix, thereby inhibiting an ordered transformation in high silicon electrical steel. This will also cause strong lattice distortion when Nb atoms enter the lattice of the Fe-Si matrix. As a result, the addition of a small amount of Nb element can significantly improve the plasticity of high silicon electrical steel [82,83].

### 5.5. Influence of Sb, Sn, and P Content

Sb, Sn, and P tend to segregate to grain boundaries. Sb may lead to a higher residual induction and a lower coercive force [84]. Shimanaka et al. [85] reported that Sb may improve the {100}<0vw> texture. Lyudkovsky [86] observed that Sb may promote {110} and {100} texture at the expense of {111} and {211}. Vodopivec [84,87] demonstrated that Sb significantly decreases {111} texture. There are two hypotheses that explain the beneficial effect of Sb during the final recrystallization process. One states that the mobility of the {111} grain boundaries decreases with Sb segregation [86]. The second theory asserts that the formation of {111} nuclei are delayed in the recrystallization process by Sb [85].

Some studies have confirmed that Sn has a similar effect on the texture of the product as Sb. During the recrystallization process, Sn segregates to the grain surface. This segregation may selectively decrease the surface energy and the mobility of some grain boundaries [88,89]. Furthermore, both Sb and Sn have been shown to protect semi-processed steel from internal oxidation during final annealing by the customer and improve the magnetic properties of fully processed cold-rolled non-grain-oriented (FP CRNO) steel.

It has been reported that P additions impede {554}<225> texture [90] and {111}<112> texture [91], and may also decrease the grain size before the cold-rolling process [90]. Other studies have also been conducted on the influence of different P contents in electrical steels [90–93]. Samples with 0.099 wt.% P exhibited better magnetic properties than 0.013 wt.% P samples. P is often used to increase hardness for improved punchability and is a major contributor to resistivity.

Furthermore, P additions also have an effect on electrical resistivity ($\rho$) and dynamic viscosity ($\nu$), in which the electrical resistivity will influence the eddy current loss and the viscosity will influence the pouring and casting process of the electrical steel. Gui et al. [94] reported that with P additions, the liquid viscosity and electrical resistivity were both increased, as shown in Figure 11. Although the properties were measured at higher temperatures, it indicates the effect of P in the alloy.

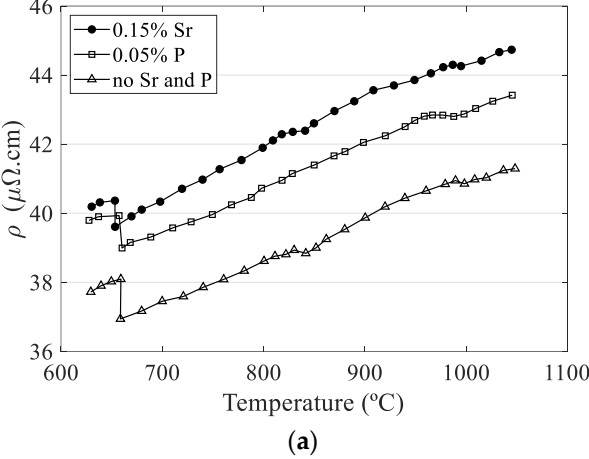
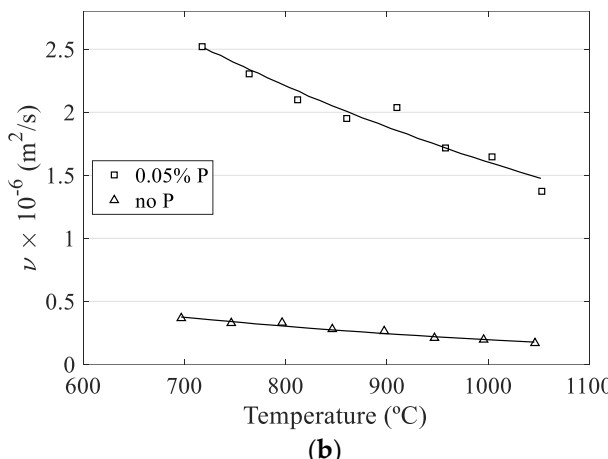

**Figure 11.** (**a**) Electrical resistivity $\rho$ and (**b**) dynamic viscosity $\nu$ of the metal at higher temperatures with and without phosphorus [94].

## 6. Typical Chemistries and Processing Paths

Efforts to achieve better magnetic properties through the influence of chemistry and processing paths for NGO electrical steels have been underway for many years.

De Dafe [58] studied the conventional processing path (thick slab casting, slab reheating, hot-rolling, cold-rolling, and final annealing) for NGO electrical steels (3.0 wt.% Si, 0.004 wt.% C, and 0.55 wt.% Mn). Samples were tested by hot-rolling at different temperatures, cold-rolling by different strains, and finally annealed at 1020 °C. De Dafe [58] finally reported that the best magnetic properties can be achieved by hot-rolling at 1000 °C and cold-rolling with a 64.3% reduction. At the final thickness of 0.5 mm, the tested $P_{15/60}$ (core loss at a condition with induction 1.5 T and frequency 60 Hz) was less than 3.0 W/kg.

He S et al. [87] also reported the application of a CaO and $CaF_2$ slag desulfurization process in steelmaking to decrease the influence of S. The NGO electrical steel (3 wt.% Si, 0.6 wt.% Al, C < 30 ppm, S < 50 ppm, N < 40 ppm), hot-rolled strips were normalized at 900~1000 °C for 2~5 min. Then, they were cold-rolled 70–80% and annealed at 1000~1100 °C for 1~2 min. The $P_{15/50}$ was reported to be less than 2.5 W/kg. Using 2-stage annealing, core loss was further decreased. The material was first annealed at 850~1000 °C for 1~3 min, and then annealed again at 1000~1100 °C for less than 1 min, with a reported $P_{15/50}$ of 2.30 W/kg.

With 1~1.3 wt.% Al, it is easier to get a larger grain size. This high Al is also helpful to avoid the harmful influence from the elements Ti, Zr, Cr, and V. The normalized 2 mm thickness strips were single stage cold-rolled to 0.5 mm. Then, they were annealed at 1050 °C for a short time. The final average grain sizes were 110~140 μm, and the surface grain size (surface to 80 μm depth) was greater than 30 μm. The final $P_{15/50}$ was 2.5~2.37 W/kg, and $B_{30}$ was 1.68 T [84].

NGO electrical steels can also be produced by a twin-roll strip-casting process, providing opportunities to process steels with higher Si contents than conventional processing allows. Some favorable textures can be generated in the as-cast condition [85,86]. Yonamine studied the strip-casting process using the directionally solidified method. Therein, a large initial columnar grain structure for 3 wt.% Si steels in the as-cast condition was observed, along with a desirable {001}<0vw> texture for electric motors [88–91]. Because the ideal texture in the as-cast condition is altered through the rolling and annealing processes, texture control in thermomechanical processing is still of great importance. De Dafe [58] studied the production of 2.75 wt.% Si NGO electrical steels by twin-roll strip-casting process. After hot-rolling, strips were rapidly cooled to a temperature below 540 °C. Then, they were cold-rolled to 0.45 mm, and finally annealed at 843 °C for 60 min. The resulting $P_{15/60}$ was 4.8 W/kg.

In the past few years, with the rapid development of electromobile market (high speed motors), the importance of NGO electrical steel strength, heat conduction, and magnetic properties under high frequencies (400–10,000 Hz) has increased. Many new works have also been undertaken to achieve better magnetic properties under high frequency conditions.

Yu Lei et al. [83] reported the application of Nb to improve NGO electrical steel strength without sacrificing magnetism. With lower temperature partial recrystallization annealing to the 0.2 mm thick cold-rolled strip, a good strength without sacrificing magnetism was reported. The $B_{50}$ is 1.572 T, $P_{1.0/400}$ is 33.26 W/kg, yield strength about 600 MPa. This strength is attributed by multiple strengthening mechanisms including dislocation, precipitation, and grain refinement strengthening.

Gervasyeva, I. V. et al. [95] studied strip thickness (0.20~0.35 mm) and cold-rolling process (single rolling ~ double rolling) influence on texture distribution and magnetic properties. As a result, the $B_{50}$ is 1.62~1.64 T, $P_{1.0/400}$ is 13.2~17.8 W/kg.

In the industrial production field, BAOSTEEL reported [96] to have ultra-thin NGO electrical steel strip products for high speed motors. The thickness of these products is 0.35–0.2 mm, for $P_{1.0/400}$ 13.0–21.0 W/kg, for $P_{1.0/800}$ 33.4–56.1, and yield strength 380–415 MPa.

## 7. Conclusions

NGO electrical steels have been developed and researched for more than 100 years, during which time a large number of reviews and books on such materials have been published. In this work, factors that can influence the magnetic properties of NGO electrical steels have been performed. Those magnetic properties are influenced not only by composition but also by final texture, grain structure, inclusion distribution, and so on.

Modern NGO electrical steels have long had much better properties. Due to differences regarding their application and markets, manufacturers are currently more interested in the strength and magnetic properties under high frequency of NGO electrical steels. This situation presents new challenges about texture control during the production process.

We already know that the texture distribution is influenced by steels chemical composition, phase transformation, solidification cooling rates, directional solidification, recrystallization annealing, deformation regime, and so on. However, texture evolution control during casting and heat treatment processes is complex, and the mechanisms involved must continue to be addressed in future research, particularly for new manufacturing processing pathways such as strip casting.

**Author Contributions:** Conceptualization, Y.D. and R.O.; methodology, Y.D.; formal analysis, Y.D. and M.F.B.; investigation, Y.D.; resources, R.O. and M.F.B.; writing—original draft preparation, Y.D.; writing—review and editing, Y.D. and M.F.B.; visualization, Y.D. and M.F.B.; supervision, M.F.B.; project administration, R.O.; funding acquisition, R.O. All authors have read and agreed to the published version of the manuscript.

**Funding:** This research was funded by Nucor Corp. and Castrip.

**Acknowledgments:** The present work is supported by Nucor Corporation. The research was performed at Missouri University of Science and Technology (Missouri S&T). All faculty and industry mentoring committee members that assisted in this work are greatly acknowledged.

**Conflicts of Interest:** The authors declare no conflict of interest.

## Nomenclature

| | |
|---|---|
| $P_{tot}$ | core loss |
| $P_a$ | anomalous loss |
| $P_e$ | eddy current loss |
| $P_h$ | hysteresis loss |
| $c$ | experimentally determined constant |
| $d$ | grain size |
| $B_{max}$ | peak magnetic induction |
| $t$ | sample thickness |
| $D$ | material density |
| $f$ | working frequency |
| $\rho$ | material resistivity |
| $k_e$ | parameters of eddy current loss |
| $k_a$ | parameters of anomalous loss |
| $k_h$ | parameters of hysteresis loss |
| $B$ or $B_H$ | magnetic induction |
| $H$ | magnetic field strength |

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
