# Peer review of "Review of Magnetic Properties and Texture Evolution in Non-Oriented Electrical Steels"

_applsci, doi:10.3390/app13106097_

Round 1
Author Response
Dear Reviewer, please see the attached document.
Regards.

Reviewer 2 Report
The paper, entitled "Review of magnetic properties and texture evolution in non-oriented electrical steels," presents some results for non-oriented electrical steel (NO). Although it is a review article, the work is not a significant contribution to the field of materials science. The article is not well written and contains very general statements as well as basic facts (at the high school level) .
I do not recommend the article for publication.
Only my selected comments can be seen below. All the mentioned comments show that the first author is at the beginning of his scientific career and should read much more scientific articles than those presented in this work.
1. Keywords: magnetic properties. I do not see magnetic properties for NGO electrical steels throughout the article. These values should be added.
2. Chapter 2 presents well-known facts from the basics about magnetism. In addition, a sentence such as „Sometimes, permeability is measured from the origin to the target location.” (line 57) is unclear.
3. The sentence „In studies addressing the magnetic properties of non-oriented electrical steels, core loss, coercive field, permeability, and magnetic induction are the main properties that dictate the magnetic material behavior” – lines 62-64, without data in the next chapters is false.
4. Strukturbericht symbols (line 149) should be replaced by structural report symbols.
5. Ref. 9 is not properly cited. This should be checked.
6. The quality of the figures is very poor.
7. What does „the final magnetic properties” mean in the following sentence „Final magnetic properties are directly effect by the texture distribution [36].” (line 190)?
8. Lines 330-331: "Hou and Liao have observed that Ce also affects the texture evolution of the material [19]." It should be explained in the paper.
9. Line 333-334: "Sb may lead to a higher residual induction and a lower coercive force [75]". It should be explained in the paper.
10. The conclusions should be rewritten.
11. The literature review is very poor, and despite rearranging 92 items, it was not selected correctly. In addition, the chaotic and incorrect style and formatting of the literature items show that the authors do not pay much attention to the works cited. It is necessary to keep the standard implemented by the journal (e.g. refs 10, 42, 58, 67).
Author Response

(The authors gave the same response as above.)

Reviewer 3 Report
The paper “Review of magnetic properties and texture evolution in non-oriented electrical steels” considers an important problem - the formation of properties in non-oriented Fe-Si alloys. Research and reviews on this topic are especially relevant now, when electric transport is rapidly developing and is replacing ICE vehicles everywhere. Non-oriented electrical steel is an important material, the improvement of the properties of which leads to a significant increase in the efficiency and energy saving of electric motors. In general, this paper may be interesting and useful for readers, but not in present form. It should be revised before publication. Here is the list of minor and major remarks:
Minor remarks:
1) “Some studies define permeability as the saturation induction because many machines run at or near saturation [1].”
How is it possible? Permeability and induction are absolutely different properties with different nature, how to define permeability as the saturation induction?
2) “However, a larger grain size also leads to a larger domain size, which in turn increases the eddy current loss [11]”
The main reason of the increase of eddy current losses in large grains is actually eddy currents, which easily occur within large grains. Domain size affects on the anomalous loss mainly.
3) “Higher Si content increases the material resistivity, which leads to a decrease in the eddy current loss. Magnetocrystalline anisotropy is the property that defines when ferromagnetic material takes more energy to magnetize in some directions more than in others. Decreasing magnetocrystalline anisotropy leads to higher permeability. Magneto striction and saturation induction are also lower when the Si content increases.”
The logical chain is broken, first the statement that silicon reduces eddy current loss is made, then general information about magnetocrystalline anisotropy without information how Si content affects on it.
4) Please present information in Fig.6a as Table, not as Figure.
5) “For instance, 1.85 wt.% Si steels have an optimum grain size of approximately 100 mm, and 3.2 wt.% Si steels have an optimum grain size of approximately 150 mm [19].”
Most likely μm should be here.
6) “The texture of electrical steels is influenced by the following factors: (1) steels chemical composition; (2) possible γ to α-phase transitions during cooling; (3) high solidification cooling rates; (4) directional solidification; (5) recrystallization annealing after the cold rolling process; and (6) magnetic annealing [1].“
In this list one extremely important factor that affects the texture is absent, namely deformation regime. At the same time the effect of magnetic annealing on the texture is questionable. The authors refer to review 10.1016/j.jmmm.2019.02.089, which, in turn, provides references to works from the 1950s, where the effect of a magnetic field on texture is not directly studied. Please provide more relevant references.
7) For electrical steels, λ-fiber is the ideal texture for good magnetic properties because it has the best magnetization direction <001> in the plane of the sheet. In general, any {hkl} with <001> axes in the plane of the sheet are preferred.
A clarification is needed. This is true only when material used in a rotating magnetic field (electric motors). For magnetic circuits, the best texture is η-fiber, since this makes it possible to orient the magnetic flux along the <100> direction for almost its entire path.
8) What kind of ODF is used, proposed by Bunge or by Roe?
9) Judging by the link [1] in the description of Figure 10, it was taken from article 10.1016/j.jmmm.2019.02.089. However, this paper does not contain this figure.
Major remarks:
1) The main problem of the paper is that it largely duplicates the previous review on the Fe-Si alloys (10.1016/j.jmmm.2019.02.089, number 1 in references list). The authors cite this paper 17 times in the text. Some figures are the same and some text is duplicated. The authors tried to pay more attention to the issues of texture formation in non-oriented electrical steels, but this part should be expanded. In this form, this review is practically useless Electrical steels have been developed and researched for more than 100 years, during this time a large number of reviews and books on such materials have been published. In order to be in demand and compare favorably with previous reviews on this topic, the article should pay increased attention to new results. A quick web of science search reveals a large number of articles published over the past few years (since 2020) describing new results on this topic, but they are not mentioned in the review. Here are just a few of them (doi):
10.3390/ma14226822
10.1016/j.actamat.2019.12.024
10.3390/ma14226893
10.1134/S0031918X20070030
10.11900/0412.1961.2019.00314
10.1016/j.matlet.2019.126844
10.3390/met12020354
2) The remark is essentially a continuation of the previous one. Due to the fact that outdated sources are used for the review, authors often indicate values of P15/50 or P15/50 (Lines 271, 363, 368 etc.) which range from 2.5 to 5 W/kg. However, modern NGO electrical steels have long had much better properties. This can be seen in the steel specification on the website of any manufacturer around the world. Of course, information from old sources can still be valuable if the goal is to trace the development of materials over a period of time. However, this was not done in the present review.
It is necessary to rework the review (at least part about texture formation) to make the paper valuable for readers. In present form in can not be published.
Author Response

(The authors gave the same response as above.)

Round 2
Reviewer 2 Report
The authors only corrected some of my comments relating to the text of the manuscript, but in my opinion the article still belongs to the group of popular science articles. Moreover, the quality of the figures is very poor (see Fig. 6). I personally do not see an improvement in the quality of the drawings, as the authors suggest. This needs to be corrected. I suggest that the authors redraw the drawings cited in the paper.
Some selected comments:
- The article contains two mathematical formulas numbered 5. Figure 9 is from ref. 5. The second equation (5) is not from ref. 9 as the authors suggest. What is the physical meaning of the Sieq parameter? BH is not defined in the text of manuscript. How can BH is related to Sieq and the material parameter p3?
- Fig. 4 was only partially copied from ref. 21. Why was the B2 phase omitted if it is described in the figure caption?
- The literature list contains a very large number of errors and needs to be corrected. Some of the cited papers include DOI, while the rest do not. All items should include all authors - please add a list of all co-authors instead of writing et al.. Please rewrite the following literature item in correct form: D. Stojakovic, R. D. Doherty, S. R. Kalidindi, and F. J. G. Landgraf, "Thermomechanical Processing for 625 Recovery of Desired $${{left}} $$ Fiber Texture in Electric Motor Steels," 626 Metallurgical and Materials Transactions A, vol. 39, no. 7, p. 1738, 2008. the ref. 44 "Y. Du, "Effect of rolling process on magnetic properties of Fe-3.3 wt.% Si non-oriented electrical steel," 594 Manuscript submitted for publication., 2021." should be updated (if the article was published) or remove. What do items 118 and 119 mean i.e. "M. K. Shimoyama Y, "JP, 59-015966,B," 1984" and "M. K. Shimoyama Y, "58-023410,A," 1983:, respectively?
Author Response
Dear reviewer,
Please, check the attached file.
Thanks.

Reviewer 3 Report
The authors have done extensive work on the revision of the paper. The review article became better after revision. Now it is definitely close to be published. However, it still contains some drawbacks.
1) Numbers of references in the reference list do not correspond with the numbers in the text. For example, the work of E.Gomes et al is mentioned at number [8] in the text, although it is at number [9] in the bibliography. This discrepancy accumulates and becomes larger towards the end of the text. The last reference in the text is [121], however bibliography contains [126] sources.
2) Lines 294-300. Authors added text fragment devoted to the effect of rare-earth elements. But why it in the section titled “Influence of Mn and S content” ? Rare-earth elements have their own specifics and their effect must be separated from other elements. I recommend making a separate small section devoted to this issue and expanding the available information. Or (worse) to remove this text fragment.
3) I don't really understand the need for having fig. 12. It shows effect of P on electrical resistivity at temperatures higher than 600 °C and viscosities of liquid metal. This information is of very dubious relevance in the context of this work and is rather out of scope.
Author Response
Dear reviewer,
Please find answers of your comments in the attached file.
Thanks.
